# Optimizing the Safety and Efficacy of Bio-Radiopharmaceuticals for Cancer Therapy

**DOI:** 10.3390/pharmaceutics15051378

**Published:** 2023-04-30

**Authors:** Cyprine Neba Funeh, Jessica Bridoux, Thomas Ertveldt, Timo W. M. De Groof, Dora Mugoli Chigoho, Parinaz Asiabi, Peter Covens, Matthias D’Huyvetter, Nick Devoogdt

**Affiliations:** 1Laboratory for In Vivo Cellular and Molecular Imaging, Department of Medical Imaging, Vrije Universiteit Brussel, Laarbeeklaan 103/K.001, 1090 Brussels, Belgium; cyprine.neba.funeh@vub.be (C.N.F.);; 2Laboratory for Molecular and Cellular Therapy, Vrije Universiteit Brussel, 1090 Brussels, Belgium

**Keywords:** antibody fragments, peptides, radionuclide, radiopharmaceutical, targeted radionuclide therapy, vectors, cancer, bio-vectors

## Abstract

The precise delivery of cytotoxic radiation to cancer cells through the combination of a specific targeting vector with a radionuclide for targeted radionuclide therapy (TRT) has proven valuable for cancer care. TRT is increasingly being considered a relevant treatment method in fighting micro-metastases in the case of relapsed and disseminated disease. While antibodies were the first vectors applied in TRT, increasing research data has cited antibody fragments and peptides with superior properties and thus a growing interest in application. As further studies are completed and the need for novel radiopharmaceuticals nurtures, rigorous considerations in the design, laboratory analysis, pre-clinical evaluation, and clinical translation must be considered to ensure improved safety and effectiveness. Here, we assess the status and recent development of biological-based radiopharmaceuticals, with a focus on peptides and antibody fragments. Challenges in radiopharmaceutical design range from target selection, vector design, choice of radionuclides and associated radiochemistry. Dosimetry estimation, and the assessment of mechanisms to increase tumor uptake while reducing off-target exposure are discussed.

## 1. Introduction

The goal of targeted radionuclide therapy (TRT) is to precisely deliver a toxic dose of ionizing radiation to disease sites by leveraging the specificity of biomolecules in targeting certain molecular patterns expressed by cells. In the field of oncology, this is used for curative, suppressive, or palliative outcome. Surgery and external beam radiotherapy have been used successfully in treating cancer; however, drawbacks including residual disease, relapse, and disseminated disease remain a concern [1]. Chemotherapy proved to be useful in fighting disseminated and residual diseases but suffers from chemoresistance and toxicity due to the non-specific nature of the mechanism of action. Targeted therapies soon developed as an alternative to overcome the constraints of conventional therapies. Targeted therapies are an inventive treatment strategy that incorporates a diagnostic step to determine the presence of molecular patterns of interest that ensure a given patient will benefit from the drug.

TRT is an attractive concept developed about 4 decades ago that has proven very useful as an add-on to chemotherapeutic modalities or as a preferred alternative where other modalities fail. This is especially useful in disseminated disease and relapse. TRT uses high-affinity and specific targeting vectors that can be administered either compartmentally or systemically, giving it the advantage of targeting both primary and metastasized cancer cells. For a radiopharmaceutical to be effective, a molecular target (receptor or antigen) exclusively expressed or (over)expressed on cancer cells must be identified. A biomolecule (targeting vector) with high specificity and affinity against the molecular target is then generated. Subsequently, a suitable radionuclide is selected and linked to the targeting vector using a suitable linkage chemistry that yields a stable radiopharmaceutical construct. This construct circulates, traces, and binds to its target, leading to an in situ radioactive decay that is toxic to the cancer cells [2]. Figure 1 depicts a schematic representation of TRT with key characteristics to consider for each component that constitutes a therapeutic radiopharmaceutical. Acting as a guide for a successful outcome of TRT, a radiotracer with similar pharmacokinetic properties as the treatment compound is usually administered followed by imaging to picture the specific accumulation of the compound to the intended site. This diagnostic step allows for patient selection, dose estimation, approximation of adverse events, therapy monitoring, and treatment follow-up.

With remarkable clinical outcomes, the field of TRT improved over the past two decades with five approved compounds: yttrium-90 ibritumomab (Zevalin^®^), iodine-131 toxitumomab (BEXXAR^®^), iodine-131 metuximab F(ab’)_2_ (Licartin^®^), lutetium-177 oxodotreotide (Lutathera^®^), and lutetium-177 PSMA-617 (Pluvicto™). Dozens of other radiotherapeutic compounds are in clinical development [3,4,5]. Amongst a plethora of targeting vectors studied, monoclonal antibodies (mAbs) were the first vectors to be exploited, with two compounds (Zevalin^®^ and BEXXAR^®^) approved by the FDA for the treatment of indolent B-cell lymphoma. However, a wide array of limitations has stalled the advancement and effective use of mAbs for TRT. This has opened the window for the development of more promising bio-vectors, peptides, and antibody fragments with superior properties as alternatives to mAbs. In the context of this paper, peptides and antibody fragments will be referred to as bio-vectors.

### 1.1. Types of Biological Vectors in TRT

The high affinity, specificity, and professional capacity of mAbs as inherent self-proteins that bind to pathological molecules are astonishing properties that led to their application as vectors for transporting radioactivity to cancer cells. Despite the maturity of the science behind radiopharmaceuticals, particularly vector design and linkage chemistry, mAb-radioconjugates remain largely unsuccessful in treating solid tumors; however, they have recorded successes in treating hematological tumors. The inefficiency in treating solid tumors is due to their large sizes (~150 kDa), that precipitates low penetration of the radioconjugate in disease tissues. Moreso, the presence of the Fc region in immunoglobulins G (IgG) mediates FcRn recycling of mAbs, extending their availability in circulation. Consequently, the long biological half-life of 10–21 days (depending on the isotype) results in off-target accumulation of radioactivity in highly vascularized organs and thus causes myelotoxicities [6]. These factors and others have resulted in decreased use of mAbs as vectors in treating solid tumors, with two times fewer clinical trials using radiolabeled mAbs for TRT from phase I to III conducted within the past 10 years [6]. 

To overcome the intrinsic limitations of mAbs, bio-vectors have been explored as alternatives. Focused on improved pharmacokinetics, better tissue penetration, and increased tumor-to-normal-tissue dose, these vectors proved to have superior properties over mAbs and virtually fit as perfect vectors for TRT [6]. Bio-vectors can be rapidly designed, synthesized, or expressed in microbial expression systems with high yields and low production costs. Their small sizes (peptides: 0.5–5 kDa; scaffold proteins: 2–20 kDa; antibody fragments: 12–110 kDa) enable rapid bio-distribution and rapid renal clearance (±65 kDa glomerular filtration threshold), resulting in a short biological half-life, access to cryptic and challenging epitopes, as well as deep penetration and homogenous distribution in tumors [7]. Bio-vectors lack a functional Fc fragment (Figure 2), resulting in the absence of immune cell activation (complement-dependent cytotoxicity and antibody-dependent cellular cytotoxicity), FcRn mediated recycling, and thus improved pharmacokinetics. The short half-life of bio-vectors in circulation allows the use of short-lived radionuclides (bismuth-213; t_1/2_ 46 min, astatine-211; t_1/2_ 7.2 h) in TRT, which are not commonly used with mAbs. This increases the spectrum of radionuclides that can be used in TRT. However, the rapid clearance and subsequent reabsorption of bio-vectors by the proximal tubules of the kidneys poses a risk of nephrotoxicity [8]. Moreso, rapid clearance may require higher or repeated dosing to ensure therapeutic efficacy. This has necessitated the development of different strategies that can overcome these downsides (as reviewed in [9]). Among many peptides investigated, somatostatin analogues such as octreotide and prostate-specific membrane antigen (PSMA) have been the most explored. Others include cholecystokinin, bombesin, and exendin analogues. Scaffold proteins under investigation include bicyclic peptides, cysteine knots, FN3 scaffolds, DARPins, ADAPTs, and affibodies [7,10]. Single-domain antibodies (SdAbs), F(ab’)_2_, Fab, scFv, minibodies, and diabodies (Figure 2) constitute the antibody fragments that have been studied at the pre-clinical and clinical levels for TRT with remarkable results. 

### 1.2. Development Status of Biological Vectors in TRT

In the last decade, a surge in radiolabeled bio-vectors developed for TRT was observed. Strikingly, the approvals of [^131^I]I-metuximab-F(ab’)_2_ (Licartin^®^) by the Chinese FDA for the treatment of metastatic refractory hepatocellular carcinoma, [^177^Lu]Lu-DOTATATE (Lutathera^®^) by the EMA and FDA for the treatment of metastatic and inoperable gastroenteropancreatic neuroendocrine tumors (GEP-NET), and more recently [^177^Lu]Lu-PSMA-617 (Pluvito™) for the treatment of PSMA-positive metastatic castration-resistant metastatic prostate cancer have played a pivotal role. Several compounds have been pre-clinically developed or are in development, with dozens in early-stage or advanced clinical studies. As the use of bio-vectors has matured and shown increasing evidence of safety and efficacy compared to mAb approaches, an upsurge in early investment by pharma companies has become noticeable. This raises hope for the much-needed funding for randomized clinical trials that are required to establish the safety and efficacy of bio-radiopharmaceuticals. The field of bio-radiopharmaceuticals remains promising; however, as new compounds are introduced, major limitations linger and need to be tackled: from design to clinical application and improvements in the therapeutic index. In this article, we highlight the major challenges and considerations critical in bio-radiopharmaceutical design: target selection, vector design, pre-clinical characterization, choice of radionuclides and associated linkage chemistry, dosimetry, and the assessment of mechanisms to increase tumor uptake while reducing off-target accumulation.

## 2. Target Selection

A successful TRT relies on the ability to identify a molecular target or antigen on tumor cells that will allow for a selective delivery of a cytotoxic payload to the tumor cells with minimal off-target effects. Target selection marks the first crucial step in drug development and thus if the wrong antigen is selected, it sets a precedent for a failure in the design and development of a radiopharmaceutical. Therefore, to initiate the development of a novel bio-radiopharmaceutical, an appropriate molecular target must be initially identified. The characteristics of the selected target, which we will discuss in this section, will determine the selection of other parameters (targeting vector, radionuclide, and chemistry) that will constitute the radiopharmaceutical. Hence, the critical principles to be considered when selecting a molecule as a target is to investigate both the physiological processes in which the target is involved with as well as to assess the expression of the target molecule, its function, its specificity, and its relevance in the disease process [11]. Al-Lazikani et al., 2016 reported the development of an online database (canSAR), which curates druggable targets for cancer therapy; an assembly of protein sequences; structural/genomic data; and chemical, pharmacological, and biochemical information that provides a broader view and criteria for selecting a target for drug development, thereby limiting bias and failure in drug development [11]. Other platforms include Guildify, DESEASES, DisGeNET, PHAROS, and Open Targets, which make use of different models for target identification [12]. Below are critical points to consider when selecting a molecular target for radiopharmaceutical development.

### 2.1. Antigen Specificity and Expression Pattern

An interesting target for TRT should be overexpressed on cancer cells in high copy numbers with low or no expression on normal healthy tissues. For a homogenous distribution of the radiopharmaceutical, the molecular target of interest should have a homogenous expression within the tumor tissue [13]. The molecular target must be membrane bound so that it is accessible to the targeting vector. Antigens so far targeted for TRT are either expressed on the cancer cells or in the tumor microenvironment (vasculature and fibroblast). Importantly, the targeted antigen should be expressed both on the primary tumor and metastasized cells to allow for the elimination of disseminated disease.

High expression of the target antigen is of interest; however, its specificity to the tumor cells is quite important. The specific expression of the target antigen will determine the toxicity profile of the radiopharmaceutical, which is usually evaluated during the characterization of the compound. TRT is based on radioactivity and thus if the radiopharmaceutical is delivered to the wrong site in the body, it often and usually causes unintended toxicities. Consequently, the targeted antigen should be highly specific or at least should have low expression on non-specific healthy tissues to minimize toxicities. Moreso, the pattern of expression on healthy tissues should be evaluated and understood to ensure a safe application. Commercialized bio-radiopharmaceuticals and current compounds in pre-clinical and clinical testing all target antigens (CD20, CD19, HER2, EGFR, PSMA, etc.) that are highly expressed on cancer cells with low expression on healthy tissues [14]. 

Some antigens (e.g., PSMA and CEA) are cleaved or secreted into circulation and can be detected in the blood pool. This could potentially lead to reduced targeting of tumors and unwanted radioactive exposure of normal cells. In addition, expression on normal cells and soluble antigens usually give rise to a phenomenon known as “antigen sink”, a situation in which soluble and non-specific antigen expression binds to the injected radiopharmaceutical, reducing the amount of radioactivity delivery to the targeted tumor or tissue [15]. Antigen sink often leads to myelotoxicity and toxicity to tissues exposed to radioactivity due to uptake of the radioconjugate. Different mechanisms have been investigated to overcome antigen sink. This includes pre-loading with a non-radioactive vector, compartmentalized administration, and the recently introduced pre-targeting approach that will be discussed in Section 7. Whatever the case, antigens that show a conspicuous non-specific expression and/or shedding in blood should be avoided or preventive mechanisms thought through to ensure suitability for the safe delivery of TRT.

### 2.2. Stability

The stability of the molecular target is also essential for a reliable TRT. Any enzymatic degradation of the target in biochemical or metabolic pathways can cause significant variation in the temporal pattern of uptake and retention of the radiopharmaceutical [13]. Genomic instability is one of the cardinal characteristics of cancer cells. This often develops variation in the expression patterns of different antigens and proteins on cancer cells, often with varying isoforms of a given antigen. Tumor cells use genomic instability as a tool to escape therapeutic pressures or to evade recognition by the immune system. In this light, the stability of targeted antigens for TRT must be understood. Knowledge of the antigens’ recycling capabilities, isoforms, and loss of expression after initial therapy cycle is important. For bio-vectors that require repeated administration in TRT, an antigen that is stably expressed is particularly important to ensure successful therapy.

### 2.3. Function

A target’s function is also of interest and influences the selection of the optimal radionuclide for therapy. When a cell surface receptor (target) is occupied by its ligand (radiopharmaceutical), oligomerization and internalization of the formed complex via endosomal or lysosomal routes can occur [13]. Antigens that are internalized upon binding to the radiopharmaceutical are beneficial since internalization and intracellular decay of the radionuclide have shown better efficacy than non-internalized antigens [16]. This is even more useful for targeted alpha therapy (TAT) and auger electron emitters (see Section 4) given the short pathlength of the emitted particle in tissues. On the contrary, for the somatostatin (sst) receptor that is highly expressed in neuroendocrine tumors, pre-clinical and clinical evidence has demonstrated that radiolabeled sst peptide antagonists (e.g., ^177^Lu-DOTA-JR11) that are non-internalizing show improved tumor uptake and therapy efficacy when compared to their radiolabeled agonist counterparts that are adequately internalized (e.g., ^177^Lu-DOTATATE) [17,18]. Nonetheless, in a TRT approach in which pre-targeting (Section 7) is considered, internalizing antigens should be avoided or carefully chosen since there is an increased likelihood of treatment failure.

It is increasingly clear that the best antigens for TRT are not only those that are highly expressed on cancer cells but also antigens that have a functional role in the growth and survival of the tumor (as reviewed in [14]). These antigens act as immune checkpoints or inducers of intracellular signaling pathways that confer drug resistance, development of a varied phenotype, invasion, migration, and metastasis. However, the argument that targeting antigens involved in cancer signaling yields better results in TRT (e.g., HER2) remains to be substantiated. Our hope is that more studies and clinical trials will provide data with better insight as the field of TRT advances.

## 3. Vector Design

The successful application of bio-radiopharmaceuticals is highly dependent on the targeting vector used as a backbone. The main challenge is to generate a targeting vector that will redirect the radionuclide in vivo with ideal pharmacokinetics, efficient tumor penetration, high specificity, fast blood clearance and non-immunogenicity. To achieve this, specific characteristics must be considered. Some of the key characteristics to consider when designing a targeting vector are shown in Table 1 and discussed below.

### 3.1. Binding Affinity

In general, vectors that bind the molecular target of interest with high affinities are preferred [19]. More specifically, vectors with high affinities (nanomolar range or better) are ideal since they allow the radionuclide close to the targeted antigen/cell for a sufficient time to induce the desired radiotoxicity to the cells. In the case of peptides, high-affinity endogenous ligands (or derivatives) have been used as targeting vectors. This holds especially true for G protein-coupled receptors and their endogenous ligands; some examples are the somatostatin receptor, GLP-1 receptor, cholecystokinin-2 receptor, or melanocortin-1 receptor [20,21,22,23]. However, phage display/biopanning is most often used as a selection method to obtain high affinity targeting vectors from large libraries of different antibody fragments, scaffold proteins, or peptides [24]. Depending on the desired targeting vector (e.g., peptides, scaffold proteins, and antibody fragments), different phage display libraries can be used. For antibody fragments, high-affinity vectors can be obtained from animals (e.g., rodents and camelids) immunized with the antigen followed by generation of immune phage display libraries, whereby the genes encoding for the antibody fragments of these animals are introduced into phage display vectors. Immune libraries often benefit from in vivo affinity maturation of the antibody-binding variable regions and as such provide a high chance of obtaining high-affinity antibody fragments against the antigen [25]. However, alternatives such as naïve libraries (from non-immunized animals) or synthetic libraries (based on antibody fragments) are used to select antibody fragments against specific proteins [26]. For peptide vectors or scaffold proteins, synthetic libraries are generally used since immune libraries are not an option [27]. Although obtaining high-affinity targeting vectors is sometimes more difficult with naïve or synthetic libraries due to lack of affinity maturation, affinities of vectors derived from synthetic libraries can be subsequently optimized using affinity maturation techniques such as targeted mutagenesis or next-generation sequencing of the original libraries [28]. Furthermore, generation of multivalent formats (bivalent, biparatopic, and trivalent) of antibody fragments can also increase the apparent affinity due to increased avidity if needed. In most cases, antibody fragments can be generated by recombinant engineering (i.e., scFvs, minibodies, and diabodies) or enzyme digestion (i.e., Fab and F(ab’)_2_) of existing validated mAbs. In addition to the display and library technologies described above, mAbs can also be obtained via hybridoma methodology.

### 3.2. Size of Targeting Vector

The size of the used targeting vector will have a great influence on the pharmacokinetics of the bio-radiopharmaceutical. Overall, smaller targeting vectors such as peptides, scaffold proteins, and certain antibody fragments (e.g., scFvs and sdAbs) have the advantage of deep penetration into targeted tissues and are rapidly cleared from the blood through the kidneys, resulting in a high target-to-background ratio in a matter of hours. However, there are exceptions to this generalization because mAbs often show better target-to-background ratios compared to scFvs or peptides, which often show perivascular retention, indicating that binding characteristics and antigen availability also play important roles. Furthermore, high kidney uptake of radiolabeled bio-vectors can result in unwanted renal damage, which will be discussed in more detail in Section 8. In contrast, monoclonal antibodies or larger antibody fragments show poor tissue penetration and/or long blood half-life, resulting in potential higher off-target effects [29].

### 3.3. Epitope and Functional Effect

An important aspect in the characterization process of the targeting vector is to identify whether the targeting vector competes with the binding of other therapeutic compounds that are already used as a standard of care aimed at the molecular target of interest and if combination therapy is warranted. One example is the HER2-targeting nanobody 2Rs15d, which recognizes a different epitope than trastuzumab. Upon radiolabeling with therapeutic radionuclides, the 2Rs15d sdAb has been used in combination with trastuzumab to enhance survival in tumor-bearing mice compared to mono-treatments [30]. In line with the binding epitope, it is also crucial to investigate the functional effect of the vector on the target. Depending on the target and its epitope, a vector could act as an agonist, antagonist, and/or inverse agonist. In addition, the vector could induce this effect in an orthosteric or allosteric manner [24]. Therefore, depending on the molecular target, it could be preferable to activate or inhibit protein activity and/or enhance or compete with endogenous ligand binding. In addition, these functional effects could influence the potential internalization and/or recycling of the target protein, which could result in reduced/enhanced uptake of the radionuclide in the target cells.

Based on the used targeting vector and its paratope, the binding epitope can be linear or discontinuous. For instance, mAbs and their derived fragments are generally believed to have a concave or flat paratope and target linear epitopes, while sdAbs have a convex paratope and are believed to target more discontinuous epitopes [31,32]. However, it is important to understand that this is a generalization and thus different targeting vectors can target linear or discontinuous epitopes. To identify the binding epitope, different methods can be deployed that will provide different in-depth information. Competition binding studies with other therapeutic compounds or binding studies using protein mutants are used commonly to obtain a general insight about the binding region and whether the targeting vector competes for binding with other compounds [33]. However, these studies are time-consuming and will generally only provide a rough estimation of the binding epitope. Alternatively, alanine scanning, in which one or a few adjacent amino acids are mutated to alanine residues, is a high-throughput method that allows a more in-depth determination of linear and discontinuous epitopes [28]. However, mutation of one or a few amino acids does not always lead to a loss of binding of the targeting vector, resulting in false-negative results, or could result in instability of the vector [34]. To date, structural studies such as crystallization, cryo-EM, or HDX-MS with the vector in complex with the antigen will provide the exact (or approximate) epitope [35]. Although these techniques will provide the most detailed information, obtaining these structures can still be difficult and/or expensive depending on the targeted antigen.

### 3.4. Stability

The physical, thermal, chemical, and in vivo stability are also important factors to consider when developing targeting vectors for bio-radiopharmaceuticals. These parameters can be assessed by examining peptide/protein aggregation, modification-prone residues within the amino acid sequence (oxidation, deamidation, and isomerization), or melting temperatures of the peptides/proteins. In addition, the in vivo stability of the targeting vector is an important parameter to take into consideration. This is often tested in vitro via serum stability studies [36]. Compared to antibody fragments and scaffold proteins, peptides are generally more prone to enzymatic degradation via endogenous proteases [37]. This remains a major concern because in vivo degradation of the peptide-based radiopharmaceutical will result in the inability to redirect the radionuclide to the target tissue, while metabolites will non-specifically bind to other tissues or be cleared from the body [36]. Therefore, multiple methods are under investigation, while some are already in clinical use to enhance the in vivo stability of peptide-based radiopharmaceuticals; these include integration of unnatural amino acids, PEGylation, cyclization, modification of termini, backbone modification, or introduction of alkyl linkers [36].

### 3.5. Immunogenicity

Treating patients with bio-radiopharmaceuticals can result in the generation of antidrug antibodies [38]. This is especially a concern upon repeated administration of the drug and can result in the loss of the therapeutic effect or severe adverse consequences in patients [38,39]. Therefore, it is important to investigate the immunogenicity of the bio-radiopharmaceutical. This can be performed using different types of assays including T-cell and dendritic cell activation assays, T- and B-cell epitope prediction, HLA binding assays, animal studies, and clinical studies [40]. Overall, immunogenicity is commonly observed in mAb-based radiopharmaceuticals that have long pharmacokinetics compared to bio-vectors that are cleared rapidly from circulation. Moreso, immunogenicity is primarily experienced with mAbs derived from non-human origins such as mice. To this end, fully human or humanized mAbs are currently the standard when developing therapeutic antibodies [41]. To avert this, antibody fragments such as sdAbs can reduce potential immunogenicity risks because they are significantly smaller, lack an Fc domain, share high homology with human VH fragments, and could even be humanized if necessary.

### 3.6. Production Process

Ideally, the targeting vector should be produced in a fast, high yield and in a cost-effective economical manner that can also be easily upscaled to a good manufacturing practice (GMP) setting during clinical translation. Hence, peptides are ideal as they are chemically synthesized in a fast and cheap manner [42]. In contrast, full mAbs are produced from mammalian cells, which makes the overall pre-clinical production process very costly [43]. Depending on the overall size, antibody fragments can usually be produced with high yields in bacterial or yeast production systems [43]. Although this lowers the production costs, this remains costly compared to the generation of peptide-based targeting vectors.

### 3.7. Chemical Modification

Finally, it is important to design a targeting vector that enables easy chemical modification to allow the incorporation of chelators or prosthetic groups without hampering antigen binding. This highly depends on the method of conjugation used (random vs. site-specific) as well as the amino acid sequence of the targeting vector (e.g., presence of lysines in the antigen binding regions). This will be discussed in Section 5.

One interesting group of biological vectors that has come under increasing investigation as targeting vectors for TRT is nanomaterials, which were initially developed and used to increase the dose of drug delivery to disease sites. With interesting properties such as a large surface-area-to-volume ratio, high radionuclide loading, and straightforward synthetic routes, they have been demonstrated as attractive vectors for the delivery of radioactivity in TRT. The use of nanomaterials in TRT is beyond the scope of this review but is detailed elsewhere [44,45,46].

## 4. Choice of Radionuclide

The radioactive payload of radiopharmaceuticals can be tailored to suit the needs of the intervention, be it for imaging or therapy. More specifically, each radionuclide exudes energetic particles or waves with distinct radiobiological properties, which could be beneficial in certain circumstances (Table 2). Hence, the choice of radionuclide is important in designing a successful radiopharmaceutical.

γ-emitting radionuclides are exclusively used for imaging purposes using single-photon emission computerized tomography (SPECT). γ-rays are electromagnetic waves with extremely short wavelengths (>10^18^ Hz) consisting of photons (Figure 3A). Their relatively low interaction with tissue confers them a high penetration power, rendering γ-rays useful for diagnostic purposes. On the downside, this poses a formidable challenge to maintain radiation protection, requiring shielding made from dense materials such as lead or concrete. Frequently used isotopes for clinical imaging include iodine-123 (^123^I), indium-111 (^111^In), technetium-99m (^99m^Tc), and occasionally iodine-131 (^131^I) [47,48,49].

β-particles comprise myriad ionizing particles and consist of electron (β^−^) or positron (β^+^) particles, which are emitted from the nucleus or (in the case of auger electrons) valence electrons expelled from atomic orbitals (Figure 3B–D). Despite β-particles sharing the same size, their nature of tissue interaction differs tremendously. For instance, β^+^-particles are exclusively used for imaging purposes and visualized using positron-emission tomography (PET), which relies on annihilation events. During such an annihilation event, an expulsed β^+^-particle collides with an electron, resulting in a burst of γ-radiation, which a PET scanner can detect as well as locate (Figure 3B). Examples of PET isotopes used in clinical instances are zirconium-89 (^89^Zr), gallium-68 (^68^Ga), and fluorine-18 (^18^F) [50,51,52].

β^−^-particles are used for therapy and (in contrast to α-particles) have a low linear energy transfer (LET) (0.1–2 keV/µm) and an increased tissue penetration ranging from 0.5 to 12 mm (Figure 3C) [53,54]. LET is the energy dissipated by the emitted particle per unit length along its ionizing pathway. For the same dose of radioactivity, particles with a high LET cause more biological damage than particles with a low LET. Due to increased tissue penetration, β^−^-particles are not only able to damage cells near the vicinity of the β^−^-source, but circumjacent cells are affected by its cross-firing effects as well [55]. The cytotoxic effects of β^−^-particles rely to a great extent on the production of reactive oxygen species [56]. These radicals deteriorate DNA and cellular proteins and contribute considerably to the cellular damage observed upon using ionizing radiation [57,58]. However, β^−^-emitters have drawbacks when dealing with hypoxic tumor sections. Ionizing radiation loses two-thirds of its functionality due to radio-resistance in hypoxic tumor regions [58]. This is due to the lack of oxygen, which is converted into reactive oxygen radicals upon using ionizing radiation. β^−^-sources used in clinical studies are mainly lutetium-177 (^177^Lu), terbium-161 (^161^Tb), ^131^I, and yttrium-90 (^90^Y) [54,59,60,61]. Concurrent emission of γ-waves during β^−^-decay is useful for imaging purposes upon treatment, allowing for patient follow-up and in monitoring therapy response.

Unlike β^−^-particles that originate from the nucleus, auger electrons are the results of a slingshot maneuver of valence electrons upon internal conversion (Figure 3D). In addition, the penetration range of auger electrons in tissue is smaller than 0.5 μm, necessitating the payload to be delivered on the membrane or nucleus of target cells for a cytotoxic effect [62,63]. Altogether, the LET of an auger electron is almost equivalent to that of an α-particle: these auger electrons carry up to 25 keV of energy [64]. Sources for auger electrons are thallium-201 (^201^TI), ^161^Tb, ^111^In, ^99m^Tc, gallium-67 (^67^Ga), and copper-64 (^64^Cu) [65,66]. A subset of auger electron-yielding isotopes is already used in the clinic for imaging purposes, allowing patient follow-up as well [65].

α-emitters are radionuclides that follow alpha-decay, during which an α-particle is emitted from the mother nuclide. An emitted α-particle consists of two protons and two neutrons, which is identical to a He242+-core (Figure 3E). Upon emission from the nucleus, this highly energetic He242+-core (5-8 MeV) interacts with cells and inflicts damage by inducing cell death [66]. Cells in the vicinity of the α-source suffer from its cross-firing effects, with the latter being very limited compared to nuclear β^−/+^-particles and γ-waves due to the limited penetration range (20–100 µm), resulting in a high LET (50–230 keV/μm) [55]. Considering that the average human cell has a diameter of 10–30 µm, we can deduce that the irradiated area around the source is very local and as such off-target effects are minimal. Hence, the need for intra-tumoral delivery is crucial. Unlike other ionizing radiation, ionization with α-particles does not necessitate oxygen, which makes them ideal for the treatment of hypoxic tumor sections [58,66,67,68,69]. DNA damage remains the primary modus operandi of α-particles as complex DNA double-strand breaks lead to cell death [66]. These features render α-emitters interesting elements for the therapy of both solid and hematological tumors. Due to the short range of α-particles, it was initially assumed to be of limited use in larger tumors and was primarily intended for the treatment of micro-metastases. However, Kratochwil et al. described a reduction in the tumor burden of patients with sizeable tumors (e.g., a diameter of >1 cm) [70]. Presently, α-emitters in general are gaining traction for translation to clinical practice as evidenced by ongoing clinical trials and investments in new production facilities [71,72]. α-sources used in clinical studies are thorium-227 (^227^Th), actinium-225 (^225^Ac), radium-224 (^224^Ra), radium-223 (^223^Ra), bismuth-213 (^213^Bi), lead-212 (^212^Pb), astatine-211 (^211^At), and terbium-149 (^149^Tb) [54,67,73]. Table 2 below provides different decay types and their applications.
pharmaceutics-15-01378-t002_Table 2Table 2Overview of decay types and their respective features. The table details all decay types of radionuclides and highlights their applications, potency to interact with tissue, the occurrence of decay, and commonly used isotopes in (pre-)clinical settings.Type of Particles/WavesApplication in OncologyPenetration Range in TissueParticle/Wave ConsistencyCommonly Used IsotopesAlpha, αTherapy20 to 100 µm [55]2 protons and 2 neutrons(He2+24)^227^Th, ^225^Ac, ^224^Ra, ^223^Ra, ^213^Bi, ^212^Pb, ^211^At, and ^149^TbBeta^−^, β^−^Therapy0.5 to 12 mm [53,54]Electron (e−10)^177^Lu, ^161^Tb, ^131^I, and ^90^YAuger, AETherapy<0.5 μm [65]Electron (e−10)^201^TI, ^161^Tb, ^111^In, ^99m^Tc, ^67^Ga, and ^64^CuBeta^+^, β^+^Imaging0.6 mm [74]Positron (e+10)^89^Zr, ^68^Ga, ^18^F, ^124^I, and ^64^CuGamma, γImagingRequires inches of lead to be stoppedElectromagnetic wave (γ00)^131^I, ^123^I, ^111^In, ^99m^Tc and ^67^Ga


## 5. Bioconjugation Strategies

The selection of a suitable bioconjugation strategy between the vector and the desired radionuclide is important in the development process of radiopharmaceuticals. The radiolabeling procedure must meet specific requirements [75].

1.The label should be compatible with the vector. The half-life of the radionuclide should be correctly matched with the biological half-life of the vector to ensure that the intended activity is delivered to the targeted tissues. The nature of the radionuclide and chemical bonds involved also have an impact on the radiolabeling efficiency, radiolabeling conditions, shelf-life, and in vivo stability of the final radiopharmaceutical.2.The conditions of the radiolabeling reaction necessary to couple the radionuclide should not denature the vector nor impact the vector’s integrity. To this end, the modification of a vector in its framework structure or complementarity determining region (CDR) has the potential to negatively affect the affinity or in vivo behavior of the compound due to steric hindrance that might impede binding to the target. Ideally, the radiolabeling strategy should not alter the vector’s affinity and should have a minimal effect on the pharmacokinetics, bio-distribution, and immunogenicity.3.When applying high radioactive amounts, radiolysis can likely occur. This is due to direct radiation damage emanating from the direct ionization of the surrounding molecules by the emitted radiation [76]. More specifically, therapeutic radionuclides have an increased propensity to cause radiolysis since they have different emission properties, and higher dosages are often used compared to diagnostic radionuclides and thus may cause more damage to the vector.

In anticipation for clinical translation, the radiopharmaceutical should be easily produced under GMP conditions and an avenue for upscaling thought out to meet the demands for clinical trials. In this section, we will cover the different bioconjugation strategies based on the selected radionuclide, their impact on the properties of the vector, as well as the challenges of these strategies for clinical translation.

### 5.1. Site-Specific versus Random Radiolabeling Approaches

A radionuclide can be conjugated to a targeting vector through a random approach or in a site-specific manner. In a random radiolabeling approach, naturally occurring amino acids in the sequence of the vector are used to anchor the functional group of interest and the radionuclide. This strategy often results in a heterogeneous final product with different amounts of coupled moieties in different positions and therefore is referred to as “random”. Hence, each regio-isomeric product potentially has a different pharmacokinetic bio-distribution and/or biological property. In the worst case, random coupling on or near the vector’s paratope may impact its functionality and affinity [77]. When selecting a random strategy, knowledge of the amount of potential attachment sites found within the structure of the vectors and their positions is important. Any random modification potentially has an extra risk of affinity loss. Despite these disadvantages, the random approach on lysines is straightforward and is already applied to functionalize vectors in clinical trials [78]. Although this methodology is already translated for some vectors, it needs to be validated and potentially optimized for each new vector.

For clinical development, it could be preferable to obtain a single homogenous product, which can be achieved by introducing a single modification site on the vector (also referred to as “site-specific modification”). In an ideal scenario, this labeling technique allows the separation of modified and non-modified vectors to avoid competition on binding sites in vivo. This increases the apparent molar activity, which renders the reproduction of results more reliable than for random bioconjugations [79,80]. Such site-specific strategies ensure accessibility of the active site of the protein, therefore minimizing the risk of influencing the affinity. One such strategy is the engineering of vectors with a *C*- or N-terminal uncoupled cysteine [79], which a free thiol group can selectively react under mild reducing conditions with maleimide-functionalized moieties, also known as “Michael addition”. One example of a prosthetic group (PG) for site-specific vector radioiodination is maleimidoethyl 3-(guanidinomethyl)-5-iodobenzoate ([^131^I]MEGMB). This prosthetic group was reported with an anti-HER2 sdAb, offering similar tumor targeting as its randomly functionalized analogue with iso-[^125^I]SGMIB but with a 2-fold higher tumor-to-kidney ratio and a 3-fold higher tumor-to-liver ratio [81]. 

Another emerging technology for site-specific labeling is the enzyme-mediated approach. This approach uses a compatible site-specific way to functionalize or radiolabel vectors. The company NBE Therapeutics has commercialized a platform for a sortase enzyme-mediated antibody conjugation (SMAC-Technology) [82]. An example is the combination of click chemistry and the enzymatic approach to take advantage on one hand of the region and stereospecificity of enzymes to obtain a modified protein which can be stored or formulated into a kit; on the other hand, it makes use of the broad versatility of click-chemical reactions for fast radiolabeling.

The click-chemistry approach is an interesting site-specific radiolabeling strategy. This term refers to the specific ligation between two substrates in a fast and quantitative manner. The click reaction can be carried out in mild conditions and yields a product that can be easily purified [83]. One class of copper-free click-chemistry reactions are the [4 + 2] cycloadditions between an electron-rich dienophile such as a trans cyclooctene (TCO) or a bicyclononyne (BCN) and an electron-deficient diene (tetrazine), also called an inverse electron-demand Diels-Alder (IEDDA). IEDDA reactions display fast kinetic constants ranging from 1 to 10^6^ M^−1^s^−1^ [83]. In the field of radiochemistry, the fast kinetics and mild conditions for the click coupling and the ease of purifying the final product are assets for labeling.

### 5.2. Radiolabeling and Functionalization Strategies: Direct versus Indirect

For the radiolabeling to occur, the vector must have a site or a chemical function on its structure that allows bio-conjugation with the desired radionuclide or with a moiety carrying the radionuclide. A wide range of techniques are available to couple a radionuclide to a targeting vector; these range from simple one-pot reactions to chemical, engineered, or enzymatic modifications. Whether or not the vector necessitates pre-modification of its structure before bioconjugation, the protocol must consider the sensitivity of the vector to certain reaction conditions. Usually, proteinaceous vectors tolerate only small amounts of solvents, require physiological pH, and are sensitive to high temperatures. Two main categories of radiolabeling strategies include direct and indirect radiolabeling.

In a direct fashion, the radionuclide is incorporated onto the vector within a single step or a one-pot reaction. For example, the direct radioiodination via aromatic substitution on tyrosine or histidine amino acids that may naturally occur throughout the structure of the vector has been proven effective on various biomolecules such as mAbs, diabodies, and peptides. This reaction occurs in mild conditions (pH 7.5 and room temperature (RT)) and within a few minutes in good yields (70–80%) [75]. In some cases, the vector must be pre-functionalized. Radiometals such as ^177^Lu or ^225^Ac require the coordination of chelates to form metal–ligand complexes. In this case, the vector must be pre-functionalized with a chelator that will allow the formation of the complex under mild conditions. Such chelators are often conjugated on the naturally occurring lysines found within the structure of the vector [84]. In this scenario, the pre-functionalized vector is purified, stored, and aliquoted for further direct labeling. Such a strategy allows the development of radiopharmaceutical kits that can be directly radiolabeled and are ideal for clinical use and commercialization [85,86].

However, the conditions necessary for direct radiolabeling vary depending on the radionuclide of interest, and many radiolabeling strategies occur under harsh conditions. Certain conditions can be incompatible with proteins, particularly the use of high temperatures and organic solvents. Another concern can be the insufficient stability of the radiolabeled construct, leading to radiocatabolites undesirably accumulating in certain tissues. For example, the accumulation of free iodine in the thyroid and salivary glands was observed when the radionuclide iodine was bound to tyrosines using the direct radiolabeling strategy [87]. To circumvent such issues, prosthetic groups (PGs) have been developed. These PGs can withstand harsher conditions while also improving in vivo behavior. In this regard, the *N*-succinimidyl guanidinomethyl [^131^I]iodobenzoate ([^131^I]SGMIB) PG was employed to overcome the release of radiocatabolites for internalizing an anti-HER2 sdAb [88]. Here, the PG was radiolabeled in the first step, purified, and/or isolated. The purification and isolation of the radiolabeled PG usually involves reverse-phase high-performance liquid chromatography (HPLC) or solid phase extraction using small cartridges. A PG is bifunctional, signifying that one part carries the radionuclide and another site carries a reactive chemical function to conjugate to the vector in a second step. Consequently, as a radioactive intermediate is produced in the first step and coupled to the vector in the second step, this strategy is referred to as indirect radiolabeling. The PG can be coupled to naturally occurring chemical functions of the vector, but the reaction can also occur in a site-specific, controlled manner. In the latter case, the protein may also require pre-labeling modification to introduce a specific reactive function as discussed above. The choice of PG or coupling strategy will depend as well on the radionuclide of choice. Below we cover the bioconjugation strategies of the radionuclides that have attracted the most interest in recent years. These radionuclides can be sorted into two main categories: radiohalogens and radiometals.

### 5.3. Radiohalogen Chemistry

Labeling molecules with radiohalogens such as radioisotopes of iodine or ^211^At generally follows the same chemistry as for non-radioactive halogenations. These reactions consist of nucleophilic substitution in the case of halogen anions and electrophilic substitution reactions in the case of electropositive halogens. However, these reactions are heavily influenced by the method of preparation of the radionuclide and by other factors such as the half-life of the radionuclide and specific activity [89].

Similar to iodination, direct astatination by nucleophilic substitution onto tyrosines has been attempted with good incorporation yields (80–90%). However, poor in vivo stability was recorded with the tyrosine–astatine bond [90]. ^211^At must be stably coupled to the vector to avoid in vivo release of the radionuclide. To expand on the number of potential carrier compounds applicable to ^211^At, efforts have been made to develop novel astatinated PGs, allowing on one hand an improvement in the incorporation of robustness while increasing in vivo stability of the final radiopharmaceuticals on the other hand [91]. Indirect strategies such as stable astatoaryl PG have been synthesized, radiolabeled, and conjugated to vectors. The main approach developed by Zalutsky et al. uses N-succinimidyl-[^211^At]astatobenzoate ([^211^At]SAB) obtained via electrophilic astatodemetalation of the organotin precursor, which is then conjugated to the amino groups of lysine residues. This strategy was employed to radiolabel a mAb involved in clinical trials [92].

Other methods such as using aryliodonium ylides for the introduction of nucleophilic astatine have been reported to generate a wide new range of PGs [93]. Despite their wide use, PGs containing activated esters have several limitations. One main concern is the competitive hydrolysis of the NHS-ester in the aqueous condition and the pH (8.5) needed for bio-conjugation on the vector’s lysines. This leads to the production of the inactive benzoate side product and suboptimal conjugation yields. Additionally, a high concentration of vector is necessary to favor the bioconjugation reaction over the competitive hydrolysis, thereby limiting the specific activity. To overcome these limitations, ^211^At-labeled PG bioconjugation via click chemistry has also been developed [94].

### 5.4. Radiometal Chemistry

Radiometals such as ^177^Lu, ^225^Ac, and thorium-227 (^227^Th) have different characteristics compared to radiohalogens. They form complexes with ligands called chelators. The radiometal features (oxidation state, atomic number, charge, and radius) determine its preferences for certain chelators. The chelators’ properties such as the number of donor atoms, coordination number, and resulting geometry with the radiometal also have an impact on the stability of the formed complex. Therefore, the choice of chelator is important to form a kinetically inert and thermodynamically stable complex. There exist two main categories of chelators: macrocyclic and acyclic chelators. Macrocycles are cage-like cyclic structures containing several donor atoms. One of the widely used macrocyclic chelators is 1,4,7,10-tetraazacyclododecane-1,4,7,10-tetraacetic acid (DOTA). Such chelators present more rigid structures with conformational flexibility to entrap the radiometal of interest, often resulting in an improved thermodynamically stable and inert complex in vivo. However, the rigid nature of these chelators often leads to elevated kinetic barriers, requiring elevated temperature for the radiolabeling procedure. Acyclic chelators have a more open structure and less restricted bond rotation, allowing the complexation with radiometals at lower temperatures. However, some of these chelators show some degree of in vivo instability as compared with macrocyclic chelators. Therefore, in selecting a chelator for a radiometal of interest, it is important to consider the properties of the radiometal and chelators and the complex stability in vitro to avoid demetallation and transmetalation in vivo [95]. Regardless of the selected chelator, other metallic impurities will compete with the complexation of the radiometal. To avoid any impact on the radiolabeling yield, the reaction must be performed under metal-free conditions. 

^177^Lu is rapidly gaining momentum in nuclear medicine. It forms a diagnostic/therapeutic pair with the positron emitter ^68^Ga for imaging because these two radionuclides share similar chemical properties. This allows treatment in combination with high-resolution and quantitative PET imaging for patient selection and response evaluation. This combination is referred to as the theranostic approach [96]. The chemical bonds produced by the Lu^3+^ ion have a strong ionic nature, requiring negatively charged hard donor elements such as oxygen for stable coordination. Negative oxygen atoms in polycarboxylate ligands appear to have a function in providing a strong ionic connection with the ionic metallic core. Therefore, polycarboxylate ligands (DOTA, NOTA, NODAGA, DTPA, and DOTRP) have been demonstrated to be the most successful choice for developing a bifunctional chelator that allows bioconjugation on vectors for subsequent complexation of ^177^Lu with adequate stability in aqueous solution and under biological conditions [97].

Another radiometal of interest for targeted alpha therapy is ^225^Ac. To date, the chelator DOTA is commonly used for ^225^Ac-labeling of peptides, mAbs, and antibody fragments. Due to the inherent high toxicity of ^225^Ac and its decay mode, some efforts are being made to facilitate the automation of radiolabeling procedures as well as in facilitating and harmonizing quality controls of ^225^Ac-labeled pharmaceuticals [98,99]. Recently, novel chelators such as MACROPA have been reported to allow the incorporation of ^225^Ac at room temperature within 5 min, which opens new possibilities for mild radiolabeling of vectors [100]. Moreso, MACROPA has the added advantage of efficiently chelating the decay daughter isotope ^213^Bi from ^225^Ac decay, a challenge that had remained a daunting task for the existing chelators used in ^225^Ac radiolabeling.

^227^Th is another attractive alpha-emitting radionuclide that has been used for TRT. It has been conjugated to mAbs targeting PSMA, HER2, mesothelin, and CD22 by means of a bifunctional chelator such as octadentate 3,2-hydroxypyridinone (3,2-HOPO) and was shown to have therapeutic efficacy in various pre-clinical tumor models [101]. Pre-clinical successes have led to several clinical trials (NCT03724747, NCT02581878, and NCT04147819). Most of these studies have been done using mAbs as the targeting vector. This could be due to the long half-life of ^227^Th (18.7 days), making it unsuitable for use with bio-vectors that are rapidly cleared from circulation (as discussed in Section 3). 

Radium-223 (^223^Ra) dichloride remains the only clinically approved alpha-emitting radionuclide for targeted alpha therapy. Due to its natural accumulating effects in bone marrow lesions, it is approved by the FDA for the treatment of metastatic castration resistant prostate cancer [102]. However, ^223^Ra until recently lacked mechanisms for conjugating it with targeting vectors for TRT. This has only begun to show promise after the introduction of the 18-membered macrocyclic chelator MACROPA, which showed a successful chelation of ^223^Ra within 5 min at room temperature [103], raising hope for the application of ^223^Ra in TRT.

Novel chelators for radiometal-based theranostics are being developed to allow for the chelation of both diagnostic and therapeutic radiometals in mild conditions. Indeed, it is advantageous to use the same precursor (combination of chelator–linker–vector) because this allows for a more accurate and personalized dosimetry estimation for radionuclide therapy. Additionally, only one GMP-grade pre-functionalized vector needs to be developed and produced, resulting in lower costs and reduced development time. An example is the novel 3p-C-NETA chelator, which is reported to have both acyclic and cyclic chelator characteristics [104]. The cyclic components provide thermodynamic stability and rigidity to the complex, while the acyclic components of NETA can accelerate the complexation process. This strategy takes advantage of the Al^18^F method in combination with therapeutic radiometals for TRT such as ^213^Bi, ^177^Lu, or ^161^Tb. As a change in radionuclide may also induce variations in the overall bio-distribution profile, a better approach could benefit from using different radioisotopes of the same element with different properties. For example, for scandium, ^44^Sc and ^43^Sc can be used for PET and ^47^Sc for TRT. In addition, for terbium, ^152^Tb and ^155^Tb can be used for PET/SPECT imaging and ^161^Tb and ^149^Tb for TRT. This will allow for the preservation of the pharmacokinetic properties of the final radiopharmaceutical.

## 6. Dosimetry

The efficacy of a therapy and the related toxicity depend on the total mean absorbed radiation dose delivered to the tumor and healthy tissues. This physical quantity is defined as the energy absorbed per unit mass of tissue and depends on the type and energy of the particle emission, the half-life of the radionuclide, and the clearance kinetics [105,106]. Dosimetry remains one of the major challenging aspects of TRT despite recent advances in estimations.

The Medical Internal Radiation Dose Committee proposed the MIRD formalism (Equation (1)) [107] to calculate the mean absorbed dose D in a target region r_T_ from activity located in one or multiple source regions r_S_ during the entire treatment time T_D_. This requires the determination of the time-integrated activity of a given source region during the treatment time and the S-value that provides the energy absorbed in the target region per unit of disintegration in the source region.
(1)$$D\left(r_{T},T_{D}\right)=\int_{0}^{T_{D}}\dot{D}\left(r_{T},t\right)dt=\sum_{r_{s}}\int_{0}^{T_{D}}A\left(r_{s},t\right)S(r_{T}\leftarrow r_{S},t)dt$$

The MIRD formalism can however be applied at several levels: from pre-clinical—both in vitro and in vivo—to a clinical setting. Traditionally whole organs were used as source and target regions. Consequently, the organ-based MIRD dosimetry approach uses organ-level S-values, which results in a mean absorbed dose in an organ/tissue. The limitations of this approach are that it assumes a uniform distribution of activity. For this reason, many studies began to apply the MIRD formalism to sub-organs [108] or even cellular levels [109], showing the importance of accurate pre-clinical dosimetry in radiopharmaceutical design.

In an extensive review, Spoormans et al. [110] emphasized the importance of detailed absorbed dose calculations in the establishment of Tumor Control Probability (TCP) and Normal Tissue Complication Probability (NTCP) models. A direct translation of the well-established TCP models from external beam radiotherapy (EBRT) to TRT is not straightforward. TCP and NTCP models might be different due to the irradiation pattern. TRT is not characterized by an acute and homogeneous irradiation of tumors. The absorbed dose in TRT is heterogeneously distributed and occurs at a low dose rate that is variable over time. Additionally, both tumor and healthy tissue responses are expected to be different when using high LET radiation compared to the low LET X-rays in EBRT.

## 7. Optimizing Radioactivity Delivery to Tumor Cells

Like any other drug, the goal in therapy is to deliver a significant dose of the compound to the disease site to bring about an effective treatment outcome. For TRT, the delivery of a maximum amount of radioactivity to the tumor is paramount. For this to happen, it follows that a good target must be selected, the right targeting vector designed, and the appropriate radionuclide selected and linked to the targeting vector using suitable chemistry followed by proper calculation of the right dose for a safe therapy. In practice, these conditions can be marred by one variable or another, leading to challenges in delivering the desired, safe, and effective therapeutic dose to tumors. The in vivo dynamics and/or specific and non-specific accumulation of the drug are cardinal causes of the unexpected variable outcomes observed in delivering the right activity to tumors. Unlike full-sized mAbs used as vectors in TRT, bio-vectors with small sizes have been proven to have rapid tumor accumulation after injection, a short blood half-life, and rapid clearance from the body through the kidneys [111]. This provides an open window for a higher amount of radioactivity to be administered in a cycle. Higher administered activity directly means a higher amount of radioactivity delivery to tumors.

One downside of rapid blood clearance and the short biological half-life of bio-vectors is that they tend to have lower cumulative radioactivity delivery to tumors [112]. Several mechanisms have been investigated to extend the circulatory half-life of bio-vectors. One such mechanism is the use of specific linkers in bio-conjugating the vector to the radionuclide or the radiolabeled PG. One of the most common types of FDA-approved linkers is polyethene glycol (PEG) linkers, which are non-toxic, non-antigenic, non-immunogenic, and water-soluble polymers. Depending on the type of vector, PEG linkers can influence tumor uptake, kidney retention, the circulation rate, and in vivo stability [97]. Additionally, PEG linkers can improve the solubility of the vector or the radiolabeled PG. In general, hydrophilicity is an important feature of radiopharmaceuticals because increasing probe lipophilicity (e.g., with PGs) has been correlated with a higher liver uptake. PEG linkers have different sizes, which may impact the radiolabeling process. In some cases, with click chemistry and pre-targeting, the vector may hinder the reactive moiety, and PEG linkers may help overcome this steric hindrance [113]. However, results will vary based on differences between the vector and the selected chemistry. This type of linker may also be specifically designed to reduce renal retention, for example via the addition of cleavable sequences incorporated between the vector and the radionuclide [114] (see Section 8). Other technologies to finetune blood clearance of antibody fragments or scaffold proteins include recombinantly engineering an albumin binding domain on targeting vectors, using bivalent antibody fragment constructs [115], and PASylation (as reviewed elsewhere [116]). Therefore, depending on the biological half-life of the targeting vector of interest, one might consider using one of the above mechanisms to increase their circulatory half-life and consequently increasing the cumulative radioactivity delivery to tumors. However, these modifications have the potential of compromising the affinity and stability of the radioconjugate in vivo and therefore must be treated carefully.

The rapid clearance of bio-vector compounds from blood through the kidneys often increases the potential of high kidney accumulation of radioactivity due to the reabsorption of the vectors in the proximal tubules (detailed in Section 8), thus making the kidney a potential dose-limiting organ [8,117]. Hence if the kidney reabsorption of bio-vectors can be prevented, the amount of radioactivity injected in a therapy cycle can be increased, and consequently the amount of radioactivity delivery to tumors can be optimized. Various mechanisms have been investigated and applied to reduce kidney retention. One such mechanism is pre-targeting, which is an alternative TRT technique that consists of first injecting a non-labeled vector modified with a specific reactive group. After the vector has accumulated at the targeted lesion(s), the patient receives a second injection. The second injection consists of a radiolabeled PG or chelator that can recognize in vivo a specific functional group on the vector, and the unbound PG or chelator is rapidly eliminated. Pre-targeting was initially introduced to avert the myelotoxicities recorded in mAb-based radiopharmaceuticals [118,119] and later with diabody [120] to increase radioactivity delivery to targeted tumor lesions for both therapeutic and diagnostic applications. However, increasing evidence now shows its potential in avoiding the non-specific accumulation of radioactivity and the antigen sink effects [121]. Interestingly, pre-targeting is potentially also useful in preventing the high kidney retention of radioactivity observed with bio-vector-based radiopharmaceuticals. With mounting pre-clinical evidence pointing to kidneys as a potential dose-limiting organ for bio-vector radiopharmaceuticals, averting this will directly lead to increased injected activity in a cycle and consequently increased radioactivity delivery to tumors. For example, Van Duijnhoven et al. reported a >20-fold higher tumor/kidney ratio using the click-chemistry method compared to the direct labeling approach with lutetium-177 (^177^Lu) [120].

Successful pre-clinical studies have led to a couple of clinical trials with a remarkable tumor-to-blood ratio and increased tumor uptake of radioactivity compared to a direct TRT approach [122,123,124]. Despite the advantages that in vivo pre-targeting can bring, it is important to consider that this strategy still requires preparation and clinical validation of at least two different compounds as well as an extra injection procedure for the patient. Additionally, the amount of each compound as well as the injection delay must be optimized. However, it offers simplified radiochemical manufacturing and a controlled process. Different strategies for pre-targeting have been investigated or are under investigation. They include streptavidin–biotin, bispecific mAbs, oligonucleotide hybridization, bio-orthogonal click chemistry, and peptide nucleic acid (PNP) (as reviewed elsewhere [119,121,125,126]).

Another mechanism under intense investigation for increasing radioactivity delivery is compartmentalized administration. Here, the radiopharmaceutical is injected directly into the cavity of the organ containing the (resected) tumor [127]. This is especially useful for organs or tissues that are difficult for the radiopharmaceutical to reach. For example, brain tumors generally have reduced access due to low amounts of radioconjugate crossing the blood–brain barrier. Thus, this reduces the amount of radioactivity that can be delivered to the tumors when the radioconjugate is administered intravenously. Compartmentalized administration is currently being investigated in a phase I/II TRT trial (NCT00089245 and NCT01099644) for brain tumors expressing B7-H3 using ^131^I-Orbuntamab. This could potentially be applied to bio-vector-based radiopharmaceuticals to increase radioactivity delivery and avoid nephro- and myelotoxicity.

## 8. Reducing Kidney Retention

Most of the side effects due to radiopharmaceuticals are under control and reversible. However, one major organ potentially described as a dose-limiting organ is the kidney, which demands critical attention.

### 8.1. Kidney Retention

The functional unit of the kidney, called the nephron, has two components. First, the main blood-filtration element, which has a controlled and constant filtration rate, is the glomerulus. Secondly, the tubular reabsorption segment, which consists of a succession of tubules with specific reabsorption and secretion properties and of which the first segment is the proximal tubule. Low-molecular-weight (LMW, <±65 kDa) radiopharmaceuticals such as bio-vectors are cleared from the body mainly via the kidneys. Due to their small sizes, they freely pass the glomerular membrane and are often then re-absorbed in the proximal tubule. Through receptor-mediated endocytose taking place on the apical side of the proximal tubule cells (PTCs), LMW radiopharmaceuticals such as bio-vectors are internalized and retained in the PTCs, causing significant kidney retention. This persistent radioactivity levels are due to the long presence of radiometabolites in the cell after lysosomal digestion. The high signal in the kidneys has become a general issue for the use of bio-vector-based radiopharmaceuticals for both therapy and imaging. The non-specific accumulation of radioactivity in the kidney is also a limiting factor for the cumulative dose for TRT. High kidney retention also hinders the visualization in nuclear imaging of tumor uptake in the vicinity of the kidneys.

Different strategies have been explored to reduce kidney retention. The megalin/cubilin receptors on the PTCs are known to be responsible for most of the re-uptake of bio-vectors [128,129,130]. Blocking megalin/cubilin-mediated endocytosis is a prevalent strategy to limit kidney retention [131,132,133,134]. This method has been largely applied using positively charged amino acids or the plasma expander Gelofusine^®^ [135]. Other strategies are being investigated to reduce kidney retention, such as the modifications of physicochemical properties of the radiopharmaceutical, pre-targeting, or cleavable linkers (Figure 4).

### 8.2. Co-Administration of Compounds Limiting the Re-Uptake of LMW Radiopharmaceuticals

Considering that the re-uptake of bio-vector radiopharmaceuticals mainly occurs via receptor-mediated endocytosis in the PTCs, blocking these receptors is a coherent approach to reduce kidney retention. Co-administration of Gelofusine^®^ has proven to be an efficient and generic but robust strategy to reduce kidney retention of bio-vector radiopharmaceuticals [117,136,137,138]. The use of Gelofusine^®^ carries the risk of a severe anaphylactic reaction and should be considered before its administration [139]. D’Huyvetter et al. in 2014 described a significant decrease in kidney retention of ^177^Lu anti-HER2-sdAb in Wistar rats when co-injected with the plasma expander Gelofusine^®^ [140]. An analogue solution is a 50/50 mixture of two positively charged amino acids (L-lysine and L-arginine). The FDA recommends a slow infusion of L-lysine and L-arginine before, during, and after the administration of Lutathera^®^, which has become the standard clinical practice [5,141]. The amino acid mixture must be administrated at least 30 min before the treatment and continued for at least 3 h. A large amount of the solution is required to significantly reduce the signal in the kidney but poses a risk of volume overload to patients, local irritation, hyperkalemia, and/or nausea. Since vomiting occurs more than 50% of the time, a 5-HT_3_ antagonist is often administered to patients before the infusion [142,143].

Another interesting compound for co-administration is metformin, a widely used antidiabetic medication that has demonstrated a strong capacity to reduce the kidney retention of peptide radiopharmaceuticals in mice when administered both orally and intravenously [144,145]. It is of great interest to further investigate the mechanism behind this because it is not metabolized, is well tolerated, and requires a low dose. Diabetes type II is a common co-morbidity with/of cancer, making the use of antidiabetic medication even more appealing to reduce nephrotoxicity [146,147,148,149]. To reduce the kidney retention of PSMA-targeting radiopharmaceuticals, mannitol was suggested by Mateucci et al. The injection of <400 mL mannitol solution before ^68^Ga-PSMA and ^177^Lu-PSMA resulted in an important reduction in kidney retention [150].

### 8.3. Physicochemical Properties Influencing Kidney Retention

It is commonly known that the kidney retention of radiopharmaceuticals is influenced by the charge and the charge distribution of the radiotracer itself. Charged spacer moieties or replacing specifically charged amino acids have shown interesting results when it comes to reducing kidney retention [151,152,153,154,155]. Different prosthetic groups used for the radiolabeling and their positions in the structure of radiopharmaceuticals are known to influence kidney retention as well [156,157]. 

Furthermore, the type of radionuclide also has an impact on kidney retention. Radiometals show higher kidney retention compared to radiohalogens because the former is typically metabolized, leaving charged radiocatabolites trapped intracellularly (the so-called “residualizing effect”) [158,159,160,161,162,163]. Therefore, neutral radiohalogenated radiopharmaceuticals are interesting candidates to improve bio-distribution compared to compounds that are labeled with radiometals [164,165]. However, depending on the chemistry and related linkers used, radiohalogens can have a high kidney retention.

### 8.4. Cleavable Linkers

Alongside the strategies mentioned above, the intercalation of a cleavable linker between the targeting vector and the radionuclide is an interesting way to reduce kidney retention. These linkers are designed to be recognized by the brush border enzymes on the apical side of PTCs or by intracellular lysosomal enzymes. Kidney retention is then lowered because of the fast excretion of the radiolabeled catabolites in the urine after cleavage by the mentioned enzymes [166]. The application of these cleavable linkers to reduce the radioactivity levels of the kidneys has already been shown to be successful with small peptides [167]. Certain amino acid sequences in the cleavable linker will be recognized more easily by the brush border enzymes and result in a more significant reduction in the kidney retention [168,169,170]. Figure 5 shows imaging of ^67^Ga-labeled Fab fragments with different cleavable linkers.

## 9. Expert Opinion and Outlook

The use of bio-vectors as alternate targeting vectors in TRT is a fast-growing concept, and they are being increasingly regarded as efficient vectors. This has been proven through several sources of pre-clinical and clinical data. mAb-based radiopharmaceuticals as initially evaluated as vectors in TRT suffer from limitations emanating from their large sizes (150 kDa) and poor therapeutic index (especially for solid tumors) and thus have experienced a reduction in application. Bio-vectors with small sizes proved to have superior characteristics as alternate vectors for TRT. Their applications have surged for the past two decades with the approval of three compounds for TRT and several ongoing clinical trials at early and late stages. TRT is based on radioactive decay to kill cancer cells, therefore if the radiopharmaceutical is delivered to the wrong site of the body, it has the potential to cause more harm to the patient than the intended therapeutic outcome. Averting this requires a careful understanding of the principle of radiopharmaceuticals and cautiously designing compounds that can provide the safest and most effective treatment outcome. This requires crucial considerations at every stage: from design conceptualization to pre-clinical and clinical translation. We believe that the speed at which bio-radiopharmaceuticals are adopted into clinical use will be defined by how safe and effective existing and novel compounds are designed, characterized, and applied. This calls for exquisite, informed decision making on the target selection, vector design, choice of radionuclide and linkage chemistry, dosimetry, and mechanisms to reduce off-target accumulation while increasing the tumor absorbed dose.

To develop a successful novel bio-radiopharmaceutical, a suitable target must be selected. This can be achieved through a comprehensive understanding of its biology and other characteristics such as the specificity, stability, and cellular function, which will allow the development of a suitable compound for TRT. Membrane-bound antigens with minimal or no shedding in blood that are uniformly and continuously expressed in high copy numbers on primary and metastasized cancer and stromal cells should be prioritized.

In selecting the appropriate targeting vector against the chosen antigen, the main goal should be to generate a targeting vector with all ideal characteristics such as rapid pharmacokinetics, efficient tumor penetration, high specificity, fast blood clearance, and no immunogenic properties. While the described biopharmaceuticals fulfill some/most of these characteristics (Table 1), there is still no class of biopharmaceuticals that can be classified as the most ideal targeting vector. In our opinion, certain small antibody fragments such as sdAbs come closest to the ideal targeting vector. While peptides and mAbs have been proven successful, in vivo stability issues (in the case of peptides) or poor tissue penetration and slow blood clearance (for mAbs) remain major issues that are not easily resolved. In contrast, sdAbs do not have these issues, and the interest in using these targeting vectors has only increased in the last decade.

A given targeting vector and targeted antigen will require a specific radionuclide to ensure a safe and effective bio-radiopharmaceutical. For instance, a slowly penetrating carrier such as an mAb radiolabeled with a short-living radionuclide might not yield the desired outcome in the treatment/imaging of solid tumors. Hence, short-lived radionuclides are a better match with targeting moieties such as bio-vectors, which exhibit a high penetrating capacity to ensure a sufficiently high tumor-to-background ratio. This issue does not apply to radionuclides with a long half-life because accumulation in the target tissue is less time-sensitive and allows for more payload deposit in solid malignant lesions, and they could be considered to radiolabel mAb-vectors for TRT [169]. A fast-clearing compound paired with a long-lived radionuclide could limit off-target radiation given its payload remains long enough in the target tissue to properly decay and results in a sufficient tumor-to-background ratio. Exceptionally, targeting non-solid tumors residing in the bloodstream and bone marrow, e.g., lymphomas and leukemias, does not necessitate the use of fast-penetrating carriers. This is due to the ease of accessibility of such malignant cells to radiopharmaceuticals, rendering tissue penetration less relevant and a long biological half-life beneficial [172].

One way to boost the efficacy of TRT is synergizing the effect of radiation with the intrinsic ability of the immune system to kill cancer cells. The immune system has been weaponized in the past decades to target cancer cells with enormous success stories. This is evident in the use of immune checkpoint blockers and CAR-T cells as targeted cancer therapies. For TRT, an increasing number of studies have highlighted the ability of TRT in priming the immune system against cancer cells. Patel et al. demonstrated that low-dose (2.5–5 Gy) ^90^Y-NM600 therapy in combination with immune checkpoint blockers (ICBs) in a pre-clinical immunologically cold tumor in syngeneic mice resulted in a strong therapeutic efficacy: 45–66% of treated mice showed complete remission compared to 0% for ^90^Y-NM600 or ICB therapy alone [173]. Recently, we reported the effects of low- (4 Gy) and high-dose (10 Gy) therapy with ^177^Lu-anti CD20 sdAb in a B16-melanoma model transfected to express the CD20 antigen. We observed an induction of a type I interferon (IFN) gene signature and a pro-inflammatory gene signature for the high- and low-dose treatments, respectively, and there was a significant increase in activated macrophages for the high-dose TRT [174]. In another therapy study using alpha-emitting radionuclide ^225^Ac radiolabeled to a sdAb targeting CD20 antigen in a B16-melanoma mouse model, we demonstrated that targeted alpha therapy can modulate a cold tumor microenvironment to an antitumoral milieu with increased infiltration of antitumoral immune cells (including macrophages and dendritic cells) and a marked increase in specific blood cytokines [175]. These examples highlight an opportunity for optimizing the efficacy of TRT with ICBs with an avenue for combination therapy. However, more data are required to ascertain the implication of a combination therapy in enhancing the efficacy of TRT.

When developing a radiopharmaceutical, a good knowledge of the vector’s biological half-life, the amount and location of potential functionalization sites, and the chemical stability of the vector in each radiolabeling condition is important. For straightforward translation, random conjugation strategies using well-described PGs are often selected. However, novel PGs or site-specific strategies may be of interest to generate homogenous radiopharmaceuticals with improved properties. In this case, the clinical-grade availability of the PGs, novel reagents, or even the availability of novel radionuclides must be considered before clinical translation. Regardless of the selected strategy, it is of utmost importance to perform in vitro and in vivo studies to ensure that the modified vector retains its affinity and remains stable. In vitro studies usually tend to overestimate the actual stability and true assessment of the radiopharmaceutical’s metabolism, and the actual stability can only be obtained from in vivo studies [176]. In the coming years, the development of theranostic approaches using novel radionuclides with vectors will surely gain momentum toward more clinical applications with the development of novel chelators and/or automated synthesis that may allow the introduction of kit-based theranostics using vectors, facilitating radiolabeling procedures and commercialization.

The delivery of a maximum amount of radioactivity to tumor cells enough to cause a therapeutic outcome is the goal of TRT. Increasing the amount of radioactivity delivery to tumor tissues hinges on the specific and non-specific accumulation of the radiopharmaceutical. For bio-vector-based radiopharmaceuticals, the kidneys have concerns as potential dose-limiting organs. It is our opinion that the design and characterization of novel bio-vector-based radiopharmaceuticals must consider mechanisms to prevent kidney toxicity from the initial steps of development. This will allow for the design of compounds with little or no effect on kidney retention. Low off-target accumulation and reduced or low kidney accumulation means that higher levels of radioactivity can be administered in a cycle, thus creating a higher probability of increasing radioactivity delivery to tumor cells.

Kidney retention depends on the type of radionuclide and the physicochemical properties of the compound (molecular weight, charge, and solubility). Tissues with fast proliferation are more sensitive to radiation and more susceptible to acute side effects, while slowly proliferating organs such as the kidney are less sensitive to radiation (but when they are affected, it leads to chronic damage with slow recovery) [177]. All this should be considered when designing novel bio-vectors for TRT and in setting up the treatment posology.

Better estimation of the radiation exposure of normal healthy tissues (more specifically, dose-limiting organs) remains a challenge when using radiopharmaceuticals. Several strategies are being explored and applied to limit kidney and hematological toxicity. Nonetheless, an effort should be made to not only improve the bio-distribution profile of bio-radiopharmaceuticals but also to better estimate the exposure of non-targeting tissue on a micro-level. Improving micro-dosimetry will naturally allow us to better understand the limitations of certain radiopharmaceuticals and improve the rationale behind their design. Today, most of the radiation toxicity estimations are made based on external-beam radiation, which differs essentially from TRT.

Clinical trials for bio-radiopharmaceuticals usher in evidence for safety and efficacy that allow licensing for the commercialization of the product. Pre-clinical studies in animal models provide a proof of concept but do not necessarily mimic the natural human biological environment. Therefore, for a radiopharmaceutical to be approved to enter a clinical trial, efficient pre-clinical data with regards to the mechanism of action, efficacy, safety, dosimetry, toxicity, and a list of other parameters are required and must be extrapolated to humans. So often, pre-clinical data under- or overestimate the implications of radiopharmaceuticals in humans. Therefore, regulatory agencies such as the FDA and EMA have put in place strict requirements for the approval of a radiopharmaceutical compound for a clinical trial. Radiopharmaceutical scientists hence must exercise due diligence in carrying out pre-clinical studies with sound data to reduce the rate at which clinical trial applications are rejected and clinical trials fail. Clinical trials require GMP-grade compounds for testing in patients. This demands that GMP facilities produce both targeting vectors and radionuclides. These facilities are scarce and often too expensive for an already underfunded field. Thus, as the field of TRT expands, the setting up of new GMP facilities should be considered to meet the demands, which will possibly reduce production costs. The bureaucracy in the approval of radiopharmaceuticals for clinical trials needs to be re-examined, especially in Europe. Sometimes, the directives of EMA conflict with legislation and directives of national health agencies. This makes compiling a dossier for clinical trials cumbersome and causes inertia in initiating trials. Furthermore, clinical trials require financing, which is habitually expensive. For the past decades, pharma companies displayed little to no interest in investing in radiopharmaceuticals. This is beginning to change after the licensing of Pluvicto™. Therefore, investment in radiopharmaceuticals is required, especially to sponsor randomized clinical trials. This is necessary to provide broad evidence of the significance of TRT in cancer therapy, and thus drive its adoption for use as a drug of choice in cancer care.

Nonetheless, several factors remain to be addressed for existing bio-radiopharmaceuticals and in the design of novel compounds to enhance their safety and effectiveness. The molecular target selection, targeting vector, radionuclide and associated chemistry, and mechanisms to improve tumor uptake while minimizing kidney retention must be keenly selected and optimized to design bio-radiopharmaceuticals with an improved therapeutic index. Most importantly, kidney accumulation and dosimetry estimation remain volatile areas that warrant focus. The design of radiopharmaceutical compounds with efficacy and safety profiles has the potential to attract much-needed funding for the clinical and randomized trials that are necessary to establish the wide impact of TRT in cancer therapy and drive the rapid adoption of bio-radiopharmaceuticals in the clinic for cancer therapy.

## Figures and Tables

**Figure 1 pharmaceutics-15-01378-f001:**
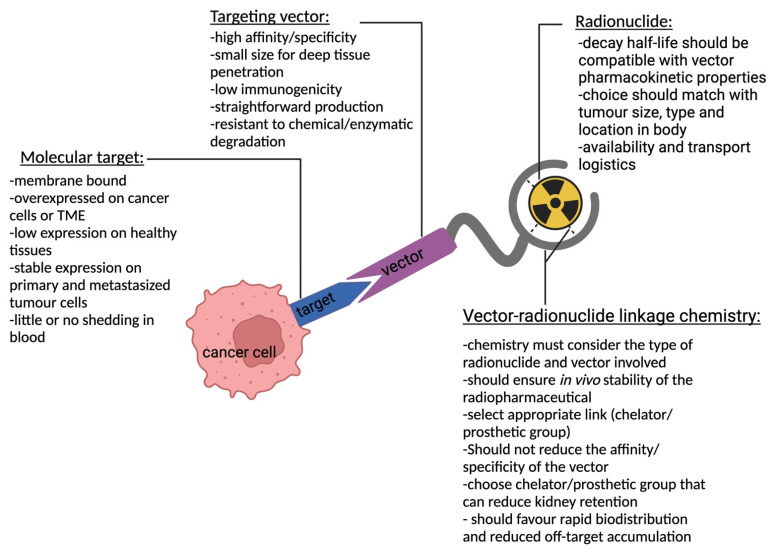
Schematic representation of TRT and key characteristics for each component depicting a targeting vector linked to a radionuclide using a chelator/prosthetic group that binds to a molecular target expressed on a cancer cell.

**Figure 2 pharmaceutics-15-01378-f002:**
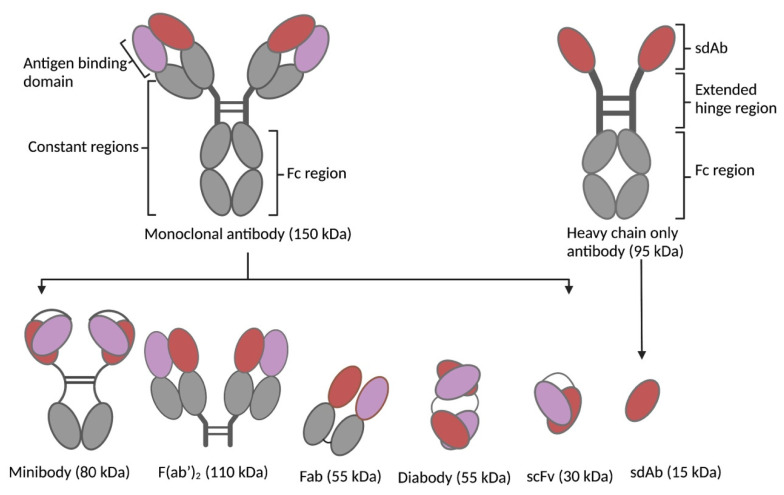
Antibodies and respective fragments investigated as targeting vectors for TRT: minibody, F(ab’)_2_, Fab, diabody, single-chain variable fragment (scFv), and SdAb.

**Figure 3 pharmaceutics-15-01378-f003:**
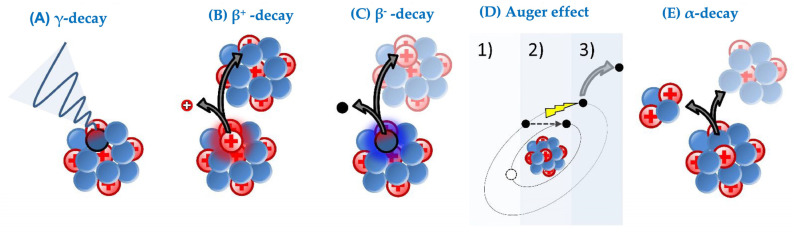
Graphical overview of various decay types and the auger effect. (**A**) γ-decay: a radioactive isotope emitting a γ-ray (blue wave) from the atomic nucleus. (**B**) β^+^-decay: a radioactive isotope emitting a positron (β^+^-particle, red dot) from the atomic nucleus and in doing so converting a proton to a neutron. (**C**) β^−^-decay: a radioactive isotope emitting an electron (β^−^-particle, black dot) from the atomic nucleus and in doing so converting a neutron to a proton. (**D_1_**) Auger effect, stage 1 of 3: an incident creates a core hole in a lower electron level. (**D_2_**) Auger effect, stage 2 of 3: an electron from a higher electron level fills up the electron vacancy by transferring to the lower electron level, dispersing energy in the process. (**D_3_**) Auger effect, stage 3 of 3: the released, transition energy is imparted onto an electron in a higher electron level, emitting it from the atom. The emitted electron is referred to as an Auger electron (AE). (**E**) α-decay: a radioactive isotope emitting an α-particle consisting of two protons (red spheres) and two neutrons (blue spheres) from the atomic nucleus.

**Figure 4 pharmaceutics-15-01378-f004:**
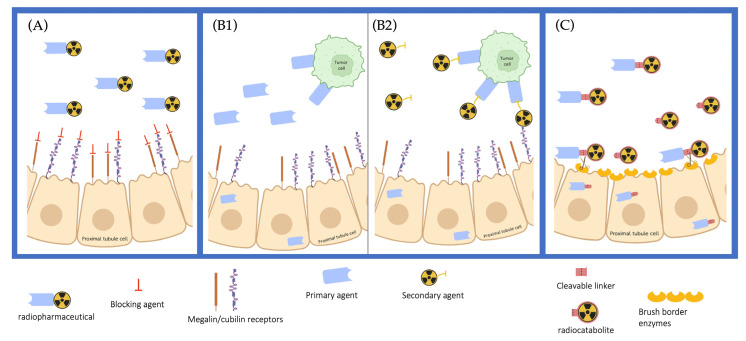
Schematic representation of the different strategies applied or under investigation to reduce the renal uptake of bio-vector radiopharmaceuticals: (**A**) blocking agents; (**B**) pre-targeting; (**C**) cleavable linkers.

**Figure 5 pharmaceutics-15-01378-f005:**
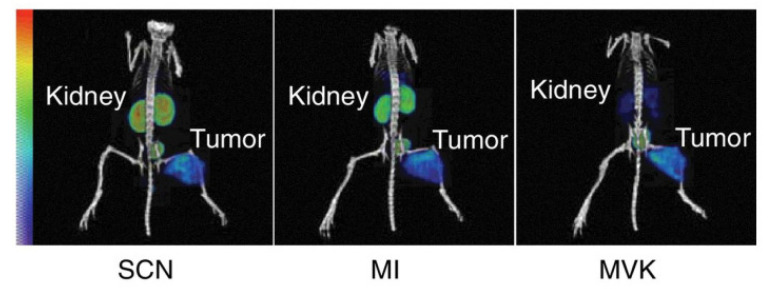
Comparative SPECT/CT images of nude mice showing tumor and kidney accumulation of radioactivity 3-h after injection of the three ^67^Ga-labeled Fab fragments, which contain cleavable linkers between the vector and chelator. [^67^Ga]Ga-NOTA-MVK-Fab provided the highest-contrast tumor image with the lowest kidney retention. Adapted from Ref. [171].

**Table 1 pharmaceutics-15-01378-t001:** Overview of characteristics of different types of targeting vectors.

Vector Characteristics	Peptides	Scaffold Proteins	Antibody Fragments	Monoclonal Antibodies
Size	0.5–5 kDa	2–20 kDa	12–110 kDa	150 kDa
Affinity	pM–μM range	pM–μM range	pM–nM range	pM–nM range
Stability	Variable	+	+	+
Tissue penetration	+	+	Low to high	-
Blood clearance Elimination route	Fast Kidneys	Fast Kidneys	Fast to slow Kidneys/liver (depending on size)	Slow Liver
Immunogenicity	-	±	-	±
Production cost	Low	High	High	Very high

## Data Availability

Not applicable.

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
