# Peer review of "Optimizing the Safety and Efficacy of Bio-Radiopharmaceuticals for Cancer Therapy"

_pharmaceutics, 2023, doi:10.3390/pharmaceutics15051378_

Round 1
Reviewer 1 Report
While this in general can be considered an interesting review on radiopharmaceuticals for cancer therapy, I would like to suggest a number of modifications.
General topics
1. The manuscript could benefit from further editing to make it more concise. A number of topics are redundant, such as 2.1 and 2.2, which are largely overlapping in content. Such redundancy should be eliminated.
2. A number of expressions appear unusual and nonscientific, such as ‘delivery of a cytotoxic punch’, ‘Targeted therapies soon develop as a silver lining’, ‘ostracizing’ and others. Did the field of TRT really ‘flourish’, when over several years only 5 compounds were approved for clinical use? Oncologists might think differently.
3. The manuscript provides no convincing reason to summarize peptides and antibody fragments as one category of targeting vectors (PAF). In fact, both text as well as table illustrate how different peptides on one hand and antibody fragments on the other are. The reviewer suggests refraining from treating them as belonging to one group and applying the unusual abbreviation PAF.
4. Chapter 3.1 completely lacks discussing natural peptide ligands of cell-surface targets such as GPCRs as a source for high-affinity vectors.
5. The manuscript briefly mentions internalization of ligands, but it should also discuss the role of agonistic (internalizing) vs antagonistic (non-internalizing) tracers.
6. Molar activity of the radiopharmaceutical as an important parameter influencing the targeting process should be discussed.
7. Chapter 9 appears rather lengthy and partially redundant, should be made much more concise.
Specifics
Fig. 1: straightforward
L167: ‘The target must be a membrane bound, that allow access for’ please rephrase
L171: Why only micro-metastasis?
L215: Ref 11 does not provide evidence for this statement.
L222: the statement about the role of the target in growth and survival appears to be speculative and needs to be substantiated by references or be deleted
L228: very general and redundant statement
L233: successful
L237: ‘To do this, specific considerations (including affinity and, stability) must 237 be considered.’ Please rephrase
Table 1: While natural peptides tend to be unstable, peptide analogs can be very stable (e.g. octreotide is ~1000-fold more stable than somatostatin). Further, a 1kDa peptide like octreotide will have a much better tissue penetration than a 110 kDa antibody fragment. The authors might want to modify the entries in the table accordingly.
L243: unsuitable reference, is a review that contributes not specifics about affinity
L282: ‘An important information in the characterization process of the targeting vector is to identify if the targeting vector competes…’ please rephrase
L298: ‘For instance, mAbs and their derived fragments are generally believed to have a concave or flat paratope and target linear epitopes, while sdAbs have a convex paratope and are believed to target more discontinuous epitopes’ please provide a reference
L328: Multiple way to improve peptide stability in vivo are not only ‘under investigation’, but have been successfully employed in compounds in the clinic and in clinical studies (e.g. SSTR and PSMA ligands).
L499: should be ‘molar activity’, see https://www.ncbi.nlm.nih.gov/pmc/articles/PMC8502193/
L787: kidney
None except for the comments on style (1.)
Reviewer 2 Report
This work is devoted to a literature review on radiopharmaceuticals for targeted radionuclide therapy (TRT). The review is well structured. The authors consistently considered the issues of target selection, vector design, choice of radionuclides, radiochemistry, and dosimetry. The work contains a small number of typographical errors, which does not affect the positive perception. However, there are a number of questions and comments presented below:
1) Table 1 lists the different types of targeting vectors. However, the authors should add the work on affibody done at Uppsala University (10.3390/cancers12030651) to this review. Affibody just occupy an intermediate position between peptides and antibody fragments in terms of molecular weight.
2) Since this work is a review, in my opinion, the authors should also mention nanomaterials as a method of delivering the radionuclide (10.1186/s12645-023-00165-y; 10.1186/s12951-019-0524-9; 10.3390/cimb44080225).
3) In table 2, in the column about Auger emitters for radiotherapy, 161Tb should be added, which is currently the most promising Auger emitter for TRT (10.2967/jnumed.120.258376).
4) Section 5.4 on isotopes for alpha therapy should be expanded. Nothing is said about radiopharmaceuticals with the thorium-227 isotope. In particular, Bayer performs several clinical trials with 227Th-labelled radiopharmaceuticals (https://clinicaltrials.bayer.com/studies/?Keyword=thorium&Latitude=&Longitude=&LocationName=&MileRadius=&page=0&SortField=Location_Distance&SortOrder=asc&Status=&ageRange=&conditions= &phases=&gender=&healthyVol=&studyType=&studyResult=&locationCountryInternal=).
In addition, nothing is said about new work related to the chelation of the radium-223 isotope, for which, in the future, it is possible to create new TRT radiopharmaceuticals (https://pubs.rsc.org/en/content/articlelanding/2021/sc/ d0sc06867e).
Round 2
Reviewer 1 Report
I fully support the publication of the manuscript in its present form.